# Fabrication and Electrical Characterization of High Aspect Ratio Through-Silicon Vias with Polyimide Liner for 3D Integration

**DOI:** 10.3390/mi13071147

**Published:** 2022-07-20

**Authors:** Xuyan Chen, Zhiming Chen, Lei Xiao, Yigang Hao, Han Wang, Yingtao Ding, Ziyue Zhang

**Affiliations:** School of Integrated Circuits and Electronics, Beijing Institute of Technology, Beijing 100081, China; 3120215377@bit.edu.cn (X.C.); czm@bit.edu.cn (Z.C.); xiaolei_1101@foxmail.com (L.X.); 3120210696@bit.edu.cn (Y.H.); 3220215100@bit.edu.cn (H.W.); ytd@bit.edu.cn (Y.D.)

**Keywords:** three-dimensional (3D) integration, high aspect ratio (HAR) through-silicon vias (TSVs), polyimide (PI) liner, Cu seed layer, redistribution layers (RDLs), S-parameters

## Abstract

High aspect ratio (HAR) through-silicon vias (TSVs) are in urgent need to achieve smaller keep-out zones (KOZs) and higher integration density for the miniaturization of high-performance three-dimensional (3D) integration of integrated circuits (IC), micro-electro-mechanical systems (MEMS), and other devices. In this study, HAR TSVs with a diameter of 11 μm and an aspect ratio of 10:1 are successfully fabricated in a low-cost process flow. Conformal polyimide (PI) liners are deposited using a vacuum-assisted spin coating technique, and the effects of spin coating time and speed on the deposition results are discussed. Then, continuous Cu seed layers are fabricated by sequential sputtering and ultrasound-assisted electroless plating. Additionally, void-free and seamless Cu conductors are formed by electroplating. Moreover, a semi-additive method is used to fabricate the redistribution layers (RDLs) on the insulating layers of photosensitive PI (PSPI). Notably, a plasma bombardment process is introduced to remove residual PSPI in the contact windows between RDLs and central pillars. Results show that the resistance of a single TSV from a daisy chain of 144 TSVs with density of 2000/mm^2^ is about 28 mΩ. Additionally, the S-parameters of a single TSV are obtained using L-2L de-embedding technology, and the experimental and simulated results agree well. The proposed low-cost fabrication technologies and the related electrical characterization of PI-TSVs are significant for the application of HAR TSVs in modern heterogeneous integration systems.

## 1. Introduction

With the continuous development of lithography technology, and manufacturing processes, novel structures such as fin-shaped field-effect transistors (Fin FETs) [1] and gate -all-around nanowire field-effect transistors (GAA nanowire FETs) [2] have been proposed to extend the Moore’s law. However, the feature size of integrated circuits (ICs) is still approaching the physical limit [3]. At the same time, as the length of the global interconnects is increasing exponentially, the delay and power consumption caused by the interconnects gradually exceeds those of the transistors and even plays a decisive role in the performance of ICs [4]. Thanks to the introduction of vertical interconnects, three-dimensional (3D) integration technology features advantages, such as reduced length and power consumption of global interconnects, increased transmission bandwidth, smaller chip footprint, and improved integration levels [5,6]. Additionally, 3D integration can realize the heterogeneous integration of multiple devices of various materials, processes, and functionalities [7,8]. As one of the key technologies for 3D integration, through-silicon vias (TSVs) can provide vertical electrical interconnects among different device layers within a compact package. In 2007, Toshiba reported the integration of TSVs in commercial CMOS image sensors, which was the first application of TSV in mass-produced devices [8]. Since then, many semiconductor manufacturers have begun to launch their own TSV products, and TSV technology has been vigorously developed and widely used. Recently, the trends in electronic products for a smaller size and better performance put forward higher requirements for TSVs, including reducing keep-out zones (KOZs), increasing interconnection density, decreasing parasitic parameters, etc. [9,10]. Therefore, the demand for high aspect ratio (HAR) TSVs with low parasitic parameters is very urgent.

Generally, the TSV usually consists of a liner, a barrier/adhesion layer, and a central conductor [11]. SiO_2_ is widely used as the liner of TSVs and traditional SiO_2_ liner is deposited by thermal oxidation or plasma-enhanced chemical vapor deposition (PECVD). However, the process temperature of thermal oxidation is too high, which is not suitable for the via-last process [12]. Additionally, the liner deposited by PECVD has many residual impurities and obvious thermal stresses [13], and it is difficult to keep continuous in HAR TSVs. Moreover, there is a huge mismatch in the coefficients of thermal expansion (CTEs) among Cu (CTE: 16.5 ppm/K), Si (CTE: 2.5 ppm/K), and SiO_2_ (CTE: 0.5 ppm/K) [14,15], which leads to severe thermal reliability issues.

Recently, polymers such as polyimide (PI), benzocyclobutene (BCB), and epoxy resin are used as liners due to their lower Young’s moduli and lower dielectric constants [16]. For example, compared with the SiO_2_ liner, PI liner can achieve smaller parasitic capacitance, smaller leakage currenst, and better thermal reliability [17]. In HAR TSVs with a small diameter, PI liner with a high step coverage has been realized by our group through a low-cost vacuum-assisted spin coating technique [18]. This approach is simple to operate, and the process temperature is relatively low (no more than 240 °C). Additionally, the fabricated PI liner can smooth the scallops caused by Bosch etching, thus avoiding interface delamination and improving the mechanical stability of the TSVs [19].

So far, there is some literature on TSVs with PI liner (PI-TSVs) [20,21,22,23]. However, most of them focus on the fabrication and tests of blind TSVs without involving double-side redistribution layers (RDLs). Therefore, it is significant to propose a whole process flow for true PI-TSVs with RDLs, especially for those with HAR. Furthermore, although the leakage current and capacitance-voltage (C-V) characteristics of PI-TSVs have been studied [24,25,26], the radio frequency (RF) performance of PI-TSVs, such as S-parameters, is still lacking in investigation.

In this paper, HAR PI-TSVs with double-side RDLs are fabricated and the electrical characteristics, including direct current (DC) resistance and S-parameters, are evaluated. A low-cost fabrication flow suitable for HAR TSVs is proposed based on the optimizations on several key processes. The effects of spin coating time and speed on the deposition results are discussed to facilitate the formation of conformal PI liners in HAR TSVs. Additionally, sequential sputtering and electroless plating is used to fabricate continuous Cu seed layers. The addition of ultrasonic assistance during electroless plating increases the deposition efficiency and improves the uniformity of the Cu seed layers. Then, void-free and seamless Cu conductors are formed by electroplating. Subsequently, photosensitive PI (PSPI) is deposited as the insulating layer in the field area. A plasma bombardment process is proposed to remove the residual PSPI in the contact windows before fabricating RDLs using the semi-additive process. The resistance of a single TSV from a daisy chain of 144 TSVs is measured. Additionally, the S-parameters of a single TSV are obtained using L-2L de-embedding technology, which are in good consistence with the simulation results.

## 2. Process Design

Figure 1 shows the schematic of the fabrication processes for the HAR PI-TSVs proposed in this paper. First, blind vias are formed by deep reactive ion etching (DRIE) (Figure 1a). Then, PI liner is deposited through a vacuum-assisted spin coating technique, as shown in Figure 1b. After that, TiN is deposited as barrier by atomic layer deposition (ALD) (Figure 1c). Next, the Cu seed layer is fabricated by sequential sputtering and electroless plating (Figure 1d), and Cu central conductors are then formed by electroplating (Figure 1e). In order to smooth the wafer surface, a chemical mechanical polishing (CMP) process is adopted to remove the redundant Cu, TiN, and PI, as shown in Figure 1f. Then, the PSPI insulating layer is deposited and patterned (Figure 1g), after which the front-side Cu RDLs are fabricated by a semi-additive process (Figure 1h). Next, the wafer is bonded upside down with a Pyrex 7740 glass holder using benzocyclobutene (BCB) as the bonding glue (Figure 1i). Subsequently, the back-side Si substrate is thinned and polished until the Cu pillars are exposed (Figure 1j). Finally, the back-side insulating layer and RDLs are also fabricated, as shown in Figure 1k,l. It can be seen that the proposed fabrication flow for the HAR PI-TSVs is low-cost, convenient, and compatible with IC processes. Actually, there are several key processes in the fabrication flow, such as the formation of liner, seed layer, and double-side RDLs, which will be discussed in detail next. Notably, in this work, some optimizations are performed on the key processes to facilitate the fabrication of the HAR PI-TSVs, including the investigation and control of spinning time and speed for the liner formation, the introduction of ultrasound assistance for the seed layer formation, the use of the PSPI insulating layer, and the proposed two-step plasma bombardment process to guarantee the connection between the Cu pillar and RDLs.

### 2.1. Vacuum-Assisted Spin Coating of PI

It is critical to deposit a conformal liner in the HAR TSVs for isolating the substrate from the central conductors and facilitating the following deposition of other layers. A low-cost and low-temperature process named the vacuum-assisted spin coating technique is used in this work to fabricate a conformal PI liner in the HAR TSVs. The process flow of the vacuum-assisted spin coating technique is shown in Figure 2, which mainly includes three steps: PI dispensing, vacuum treatment, and spin coating. Before the dispensing step, the as-etched substrate is first cleaned with acetone, isopropyl alcohol (IPA), and deionized water (DI water) with the assistance of ultrasound treatment, followed by an O_2_ plasma clean to increase its adhesivity with PI. Then, PI (PI-5J, YiDun New Material Co., Ltd., China) with a specific viscosity is dispensed from the center of the sample outwards with a dropper until the sample surface is completely covered by PI (Figure 2a). After which, the sample is immediately transferred to a vacuum chamber (HASUC, DZF-6050) for a 5 min vacuum treatment. Due to the pressure difference between the trapped air inside the blind vias and the external vacuum environment, the trapped air can escape out rapidly and the PI will fully fill the interior of the blind vias (Figure 2b). Next, the PI inside the vias is spined out except for that attached to the sidewall through a two-step spin coating procedure. The low-speed state will shake off the excess PI in the field area and level the surface, while the PI inside the blind vias will be spined out under the actions of centrifugal force and intermolecular force in the high-speed state [14], leaving a thin PI layer on the sidewall of the vias, as shown in Figure 2c. Finally, the PI liner is cured in N_2_ ambient; more details of this will be mentioned in Section 3.

### 2.2. Electroless Plating of Cu

The quality of the seed layer has a great influence on the result of upcoming Cu electroplating, and a continuous and dense seed layer is necessary to form a good central conductor. Since sputtering is not sufficient to fabricate continuous Cu seed layers in TSVs with aspect ratios exceeding 10:1, and the adhesion of electroless Cu to the TiN barrier is not so good, a combination of sequential sputtering and electroless plating is proposed to achieve continuous Cu seed layer deposition in HAR TSVs [27,28]. The electroless plating process mainly includes six steps, as shown in Figure 3. The sample is first immersed into DI water and a vacuum treatment is conducted to evacuate the air inside the vias. Then it is transferred to Conditioner solution to clean the sample surface. Immediately thereafter, the sample is placed in the pre-dip solution and another vacuum treatment is carried out to fully wet the reaction surface. Next, Pd ions are first adsorbed and then reduced on the via sidewall in the activator solution and reducer solution, respectively, which act as the catalysts to the Cu reduction. Finally, Cu seed layers can be formed by the reduction in Cu ions with the catalysis of Pd in the Electroless Cu solution. The solutions used in the electroless plating process are commercially available (Neoganth, Atotech Ltd., Germany). Notably, using ultrasonic assistance will help Cu ions to diffuse into the vias, thus enhancing the reaction efficiency and improving the step coverage of seed layers [29].

### 2.3. Fabrication of PSPI Insulating Layer and Cu RDLs

After the formation of blind TSVs, the fabrication of double-side surface insulating layers and RDLs is also crucial for the TSVs to realize vertical electrical interconnections. PSPI (JAPB-101, 1000–1100 cP) is used as the field insulating layer in this work, and its deposition process is similar to that of photoresist. Actually, there are two key points in the fabrication of PSPI insulating layer. On the one hand, the contact window on the PSPI for the electrical connection between the TSV and the RDL should be located exactly on the central pillar and avoid exposing the substrate. Ideally, the center of the window is on the same line as the axis of the TSV, as shown in Figure 4a. However, due to factors such as the error of lithography alignment, the flatness of sample surface after CMP, and the steepness of the vias caused by DRIE, there are deviations in the actual lithography pattern, as shown in Figure 4b. In severe cases as shown in Figure 4c, there will be serious leakage between the RDL and the substrate due to the exposure of surrounding substrate, which will destroy the performance of the device. Therefore, in this paper, the diameter of the contact window is designed to be 4 μm smaller than that of the central pillar for larger tolerances. On the other hand, the contact window must be clean with no PSPI residues to ensure good contact between the RDL and the pillar. In this work, a plasma bombardment treatment is introduced to guarantee the removal of PSPI in the contact window, which will be discussed in detail in Section 3.

The RDLs are fabricated by the semi-additive process, as shown in Figure 5. The steps include: (a) First, the Ti barrier layer and Cu seed layer are sputtered on the PSPI layer in sequence. (b) Next, the photoresist is deposited and patterned to restrict the area for the RDLs. (c) Cu RDLs are then fabricated by electroplating. (d) After that, the photoresist is removed, and (e) another photoresist layer is deposited and patterned to protect the RDL area. (f) Finally, the exposed Cu seed layer and Ti barrier layer outside the RDL area are etched by wet etching and the photoresist is removed to complete the RDL fabrication.

## 3. Fabrication

### 3.1. Deposition of PI Liner

First, blind vias with a diameter of 11 μm are etched by DRIE, as shown in Figure 6. The depth of the vias is about 126 μm, and the aspect ratio is as high as 11:1. Notably, the density of the blind vias is as large as 2000/mm^2^.

The morphology of the PI liner deposited by the vacuum-assisted spin coating technique is affected by many factors, such as the spinning time and the spinning speed. In order to explore the influence of these parameters on the deposition results, two groups of comparative experiments are carried out. In the first group, the spinning time of high-speed state is set to be 20 s, 40 s, and 60 s, individually, and the spinning speed is 2000 rpm. Figure 7a–c show the cross sectional SEM images of the fabricated PI liners with different spinning times. It is shown that as the spinning time increases from 20 s to 40 s, the thickness of the PI liner decreases for nearly all positions. Moreover, the uniformity of the PI liner from the top corner of the via to the near bottom is improved. This is because 20 s is too short for the PI on the via sidewall to reach a stable state between being spined out and remaining in situ. Therefore, with a longer spinning time, more PI is spined out especially for the positions where the PI layer is much thicker, and the liner becomes thinner and more conformal. However, as the driving forces such as centrifugal force, intermolecular force of PI, gravity of PI, and frictional force between PI and the via sidewall have already reach a balanced state for a spinning time of about 40 s, the PI thickness will not obviously decrease when the spinning time further increases to 60 s, as shown in Figure 7c. To evaluate the influence of the spinning speed, another PI liner is deposited with a spinning speed of 3000 rpm and a spinning time of 40 s, as shown in Figure 7d. Compared to Figure 7b, when the spinning speed increases from 2000 pm to 3000 rpm for 40 s, the PI thickness is reduced especially at the via middle as the centrifugal force is strengthened.

In this paper, the spinning parameters of the high-speed state are set to be 2000 rpm for 40 s. Additionally, the viscosity of PI is about 4000 cP, the vacuuming time is 5 min, and the low-speed spinning state is 500 rpm for 12 s. In addition, the liner is cured at 240 °C for 4 h in N_2_ ambient to fully polymerize the PI. As shown in Figure 7b, the PI liner is conformal, and the thickness is about 630–660 nm at the sidewall top, 450–460 nm at the sidewall middle, and 450–480 nm at the sidewall bottom, with a step coverage of above 15%. Moreover, the surface of PI is smooth without duplicating the sidewall scallops caused by Bosch etching. Considering that the PI layer on the top surface will be removed in further processes, the uniform and continuous PI layer at the via sidewall contributes to ensuring a good insulation characteristic of the final TSV. Notably, the highest process temperature in the vacuum-assisted spin coating technique is only 240 °C during the curing procedure, making it compatible with IC processes.

### 3.2. Deposition of Cu Seed Layer

Next, a uniform TiN barrier layer with a thickness of about 80 nm is deposited on the PI liner by ALD. Considering that the blind vias have a HAR of more than 10:1, neither conventional sputtering nor electroless plating can easily and effectively fabricate a continuous Cu seed layer in such HAR vias; therefore, a novel technique named sequential sputtering and electroless plating is used to fabricate a continuous Cu seed layers. First, sputtering is used to deposit 100 nm Cu as the pre-treatment for electroless plating. Then, electroless plating of Cu is performed to achieve a continuous seed layer inside the blind vias. Due to the limited capability of the sputtering machine (IMECAS, JS-3), a continuous Cu layer can only be realized within one-third of the depth of the blind vias, and there are only discrete Cu particles at the via middle and the via bottom after sputtering. Even so, the Cu particles are able to enhance the reaction efficiency of the upcoming electroless plating and guarantee the continuity of the seed layer especially at the via bottom [27].

During the electroless Cu step where Cu ions are reduced, the exchange rate of the solutes inside the blind vias is slow so that the growth rate of the seed layer at the via bottom is much slower than that at the via top. Moreover, the gas generated during the reduction will accumulate inside the blind vias and affect the transport of the reactants. Therefore, the electroless Cu plating will take a long time, and the deposited Cu seed layer will be thick at the via opening and thin at the via bottom without additional interference. In this work, an ultrasonic assistance is introduced to accelerate the transport of reactants, shorten the reaction time, and improve the step coverage of the Cu seed layers.

It should be noted that the seed layer is fragile at the beginning of the reaction, so there should be about 5 min before applying ultrasound assistance to protect the initial Cu layer. Additionally, the power, duration, and duty cycle of ultrasound will affect the morphology of the seed layer. In the case of low power or short duration, it is not enough to shatter the bubbles in the blind vias, and the transport of reactive substances is still limited. However, in the case of high power or long duration, it will affect the adhesion of Cu and even destroy the continuity of the seed layer. On the other hand, lower duty cycle and longer response time will bring in weaker ultrasonic intensity, which also means that the thickness of the seed layer at the field and openings will increase.

In this work, the ultrasonic power is 180 W, and it is turned on for 1 s per min. The total treatment time is 40 min. The thickness of the formed seed layer is about 500 nm at the via opening and about 40 nm at the via bottom, as shown in Figure 8. It can be seen that the Cu seed layer is continuous and dense, which can satisfy the requirements of electroplating. Cu electroplating is then conducted to fabricate central pillars, and the result is shown in Figure 9. It shows that the blind vias are fully filled by Cu with no obvious voids or seams, proving the quality of the fabricated seed layer.

### 3.3. Fabrication of PSPI Insulating Layer in Field Area

After the electroplating is completed, the Cu and PI in the field area are removed by CMP until the Si substrate is completely exposed. A PSPI layer is then coated and patterned as the surface insulating layer. As the patterning of PSPI is performed by lithography and development, there always leaves over a thin PSPI film of about 10 nm in the patterned window for the connection between the RDL and the Cu pillar, leading to the open circuit failure. Therefore, a two-step plasma bombardment process is proposed in this work to remove the residual PSPI film. First, O_2_ plasma bombardment is used to clean the PSPI in the contact window, which results in the slight oxidation of the Cu pillar. Next, the Cu oxide layer is etched by Ar plasma bombardment. To verify the feasibility of the plasma bombardment process, two identical samples with PSPI layer on Cu layer are prepared. After development, one sample is treated by the proposed two-step plasma bombardment process and the two samples are then electroplated. The electroplating results are shown in Figure 10. It can be seen that only two contact windows of the sample without bombardment treatment are able to be plated, which is due to that most of the windows containing residual PSPI films. In contrast, all contact windows are successfully plated after plasma bombardment, as shown in Figure 10b, indicating that the residual PSPI films are effectively removed. Therefore, the plasma bombardment process enables the formation of good electrical connections between RDLs and Cu pillars.

After the fabrication of RDLs by the semi-additive process, the front side of the sample is bonded with a glass slide using BCB as the bonding glue. The back side of the sample is then thinned until about 10 μm away from the bottom of the blind vias. The remaining thickness is removed by CMP to reveal the Cu pillars, which can also reduce the surface roughness and mitigate the residual stress caused by thinning [30,31]. Note that there are no cracks on the exposed Cu pillars nor obvious wafer warpage after CMP. Finally, the back-side PSPI insulating layer and RDLs are also fabricated. The final fabrication result is shown in Figure 11. The TSVs are dense and intact with double-side RDLs, showing the feasibility of the proposed process flow.

## 4. Electrical Characterizations

In this work, the DC resistance and S-parameters of the HAR PI-TSVs are tested. As shown in Figure 12, a daisy chain structure and its corresponding decoupling structure are designed to calculate the resistance of a single TSV. The daisy chain consists of 144 identical TSVs that are connected in series through RDLs. For each structure, the test voltage is applied to the probe pads by two test probes to measure its current–voltage (I–V) characteristic using Keysight B1500A. Then, the resistance of each structure is calculated according to the I–V curve. Furthermore, the resistance of a single TSV is obtained by subtracting the resistance of the decoupling structure from that of the daisy chain structure, and then dividing by 144, which is the number of the TSVs in the daisy chain structure. The I–V characteristics of TSV daisy chains with distribution density of 1000/mm^2^ and 2000/mm^2^ and the corresponding decoupling structures are evaluated, as shown in Figure 13, and the resistances of a single TSV are calculated to be about 33 mΩ and 28 mΩ, respectively. Note that all the calculated resistances from more than ten groups of test structures for each TSV density fall within the 3σ range, where σ refers to the standard deviation, showing the good stability of the fabrication flow.

To evaluate the RF performance of the PI-TSVs, the S-parameters of the TSVs are first simulated using an ANSYS High-Frequency Structure Simulator (HFSS), and the simulation structures are shown in Figure 14. First, the S-parameters of the four structures are simulated individually. Then, the S-parameters of a single TSV can be obtained by decoupling through the L-2L de-embedding method [32,33], where the S11 and the S21 at 40 GHz are −20.7 dB and −0.56 dB, respectively. The L-2L de-embedding structures are also fabricated and measured to obtain the actual S-parameters of the PI-TSV in this paper. Figure 15 plots the comparison between the simulated and measured S-parameters of a single TSV. It can be seen that the trend of the tested S-parameters after de-embedding agrees with those of the simulated ones, except for that the former ones exhibit small fluctuations due to the intrinsic fluctuations of the RF test probes at high frequencies.

## 5. Conclusions

In this paper, a low-cost and feasible process flow for HAR PI-TSVs is proposed. Conformal PI liner is deposited through a low-cost and low-temperature vacuum-assisted spin coating technique, and the influence of spinning time and speed is discussed. Additionally, a continuous and dense Cu seed layer is achieved by sequential sputtering and electroless plating with the assistance of ultrasound treatment, and then, void-free and seamless Cu pillars are formed by electroplating. Moreover, PSPI is deposited as the insulating layer in the field area and a two-step plasma bombardment process is proposed to remove the residual PSPI in the contact windows, which improves the connection quality between Cu pillars and RDLs. Then, the RDLs are fabricated by the semi-additive process. HAR PI-TSVs with a diameter of 11 μm and depth of 110 μm are successfully fabricated with double-side RDLs, and the electrical performance, including DC resistance and S-parameters, is evaluated. The resistances of a single TSV in arrays with distribution density of 1000/mm^2^ and 2000/mm^2^ are measured to be about 33 mΩ and 28 mΩ, respectively. Additionally, the test results of the S-parameters of a single TSV are in good agreement with the simulation results. The proposed fabrication scheme and the electrical characterization of HAR PI-TSVs enrich the study on TSVs with polymer liners, and facilitate the application of HAR TSVs in the miniaturization of modern high-performance heterogeneous integration systems.

## Figures and Tables

**Figure 1 micromachines-13-01147-f001:**
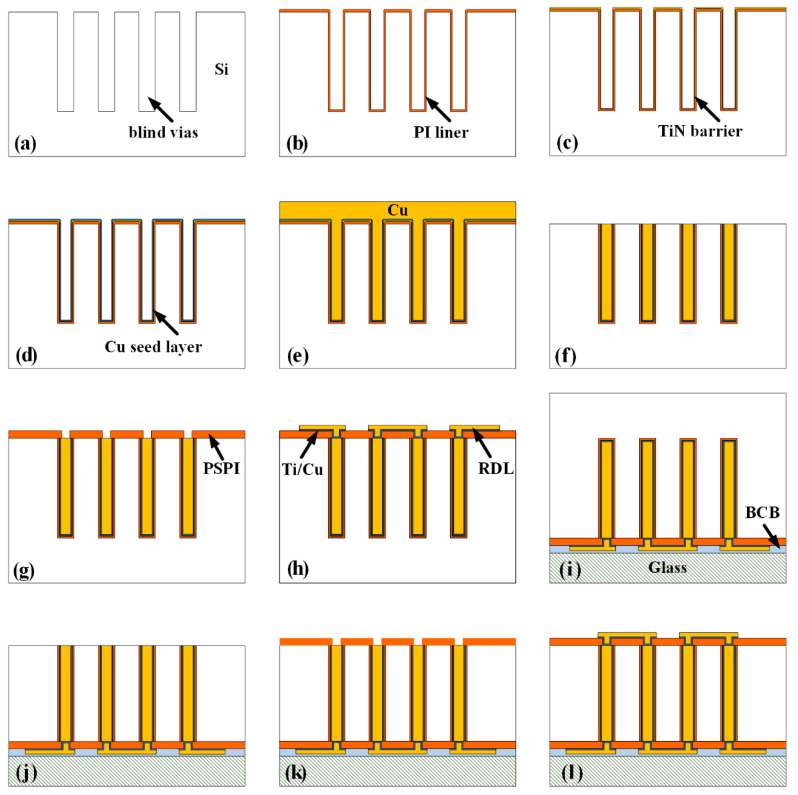
Schematic of the process flow for the HAR PI-TSVs: (**a**) Si DRIE, (**b**) vacuum-assisted spin coating of PI, (**c**) ALD of TiN, (**d**) sputtering and electroless plating of Cu seed layer, (**e**) Cu electroplating, (**f**) front-side CMP, (**g**) deposition of front-side PSPI insulating layer, (**h**) fabrication of front-side Cu RDLs, (**i**) permanent bonding, (**j**) back-side thinning and CMP, (**k**) deposition of back-side PSPI insulating layer, (**l**) fabrication of back-side Cu RDLs.

**Figure 2 micromachines-13-01147-f002:**
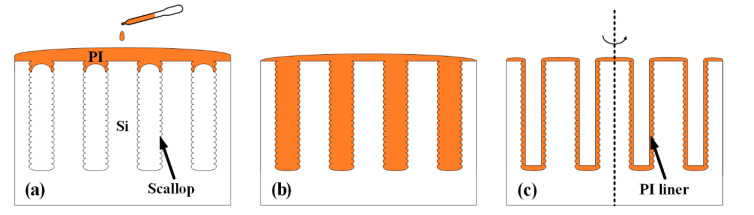
Schematic of the vacuum-assisted spin coating technique: (**a**) dispensing of PI, (**b**) vacuum treatment, (**c**) spin coating of PI.

**Figure 3 micromachines-13-01147-f003:**
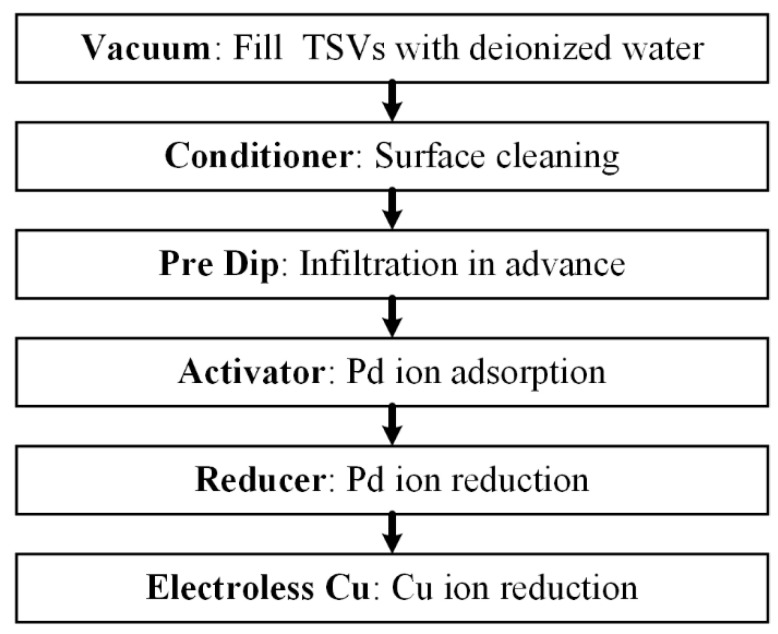
Steps of electroless plating of Cu.

**Figure 4 micromachines-13-01147-f004:**
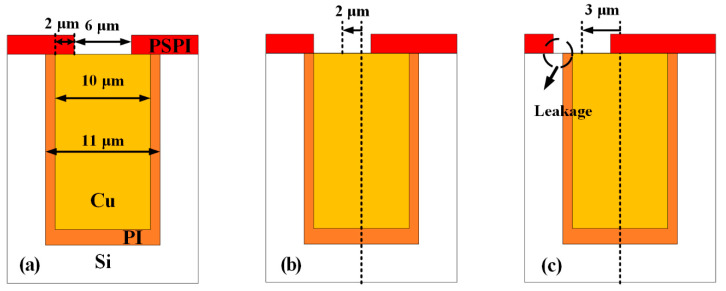
Schematic of lithography alignment of PSPI insulating layer: (**a**) ideal situation, (**b**) slight deviation, (**c**) serious deviation.

**Figure 5 micromachines-13-01147-f005:**
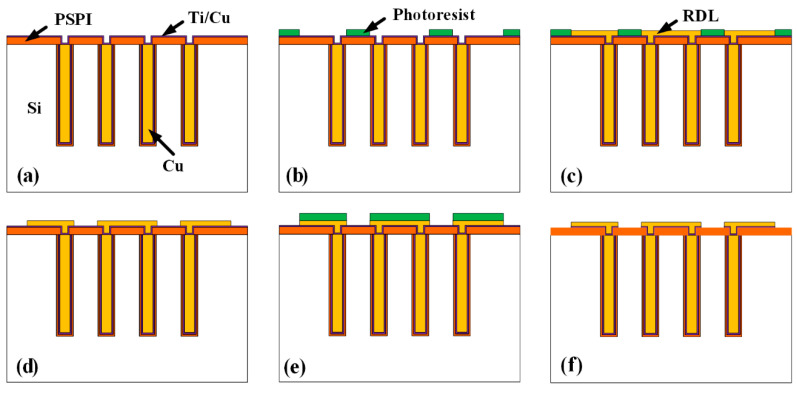
Schematic of the semi-additive process: (**a**) sputtering of barrier and seed layer, (**b**) patterning of RDL area, (**c**) RDL electroplating, (**d**) removal of photoresist, (**e**) patterning to protect RDL area, (**f**) wet etching of excess barrier and seed layer and removal of photoresist.

**Figure 6 micromachines-13-01147-f006:**
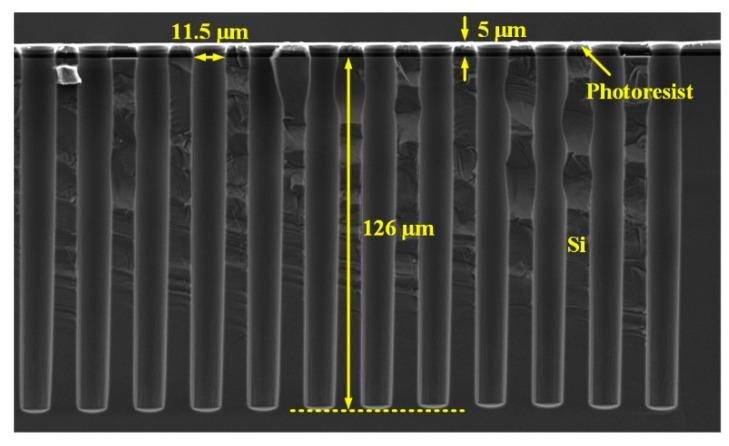
Cross sectional SEM image of the blind vias etched by DRIE.

**Figure 7 micromachines-13-01147-f007:**
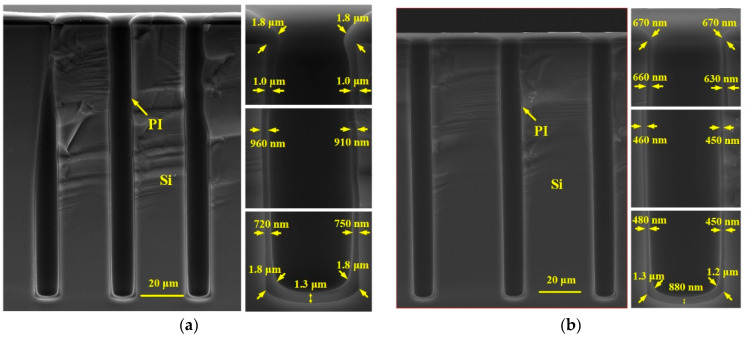
Cross sectional SEM images of PI liners deposited by vacuum-assisted spin coating technique: (**a**) 2000 rpm for 20 s, (**b**) 2000 rpm for 40 s, (**c**) 2000 rpm for 60 s, (**d**) 3000 rpm for 40 s.

**Figure 8 micromachines-13-01147-f008:**
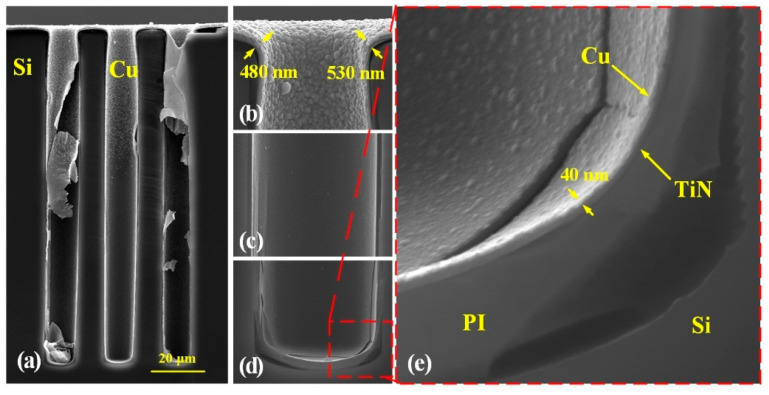
Cross sectional SEM images of fabricated Cu seed layer: (**a**) global view, (**b**) via top, (**c**) via middle, (**d**) via bottom, (**e**) enlarged view of via bottom.

**Figure 9 micromachines-13-01147-f009:**
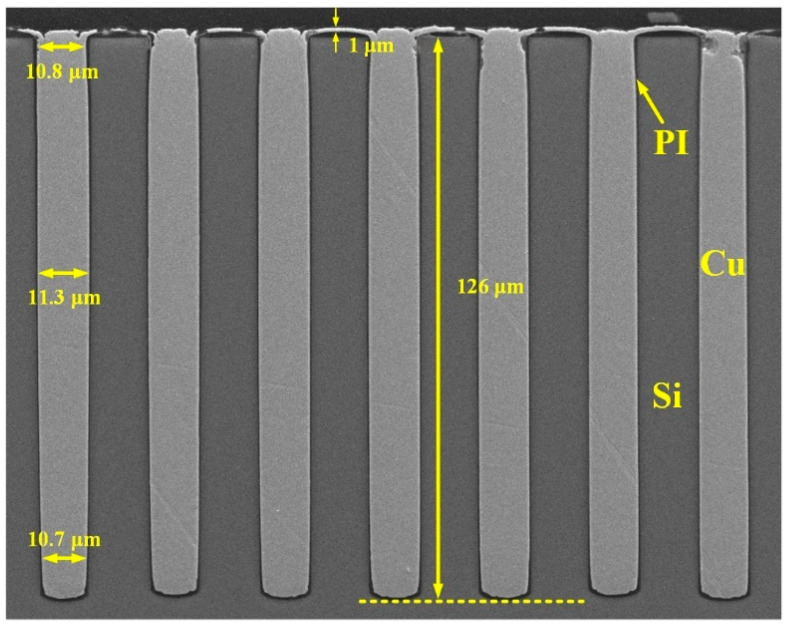
Cross sectional SEM image of blind TSVs after Cu electroplating.

**Figure 10 micromachines-13-01147-f010:**
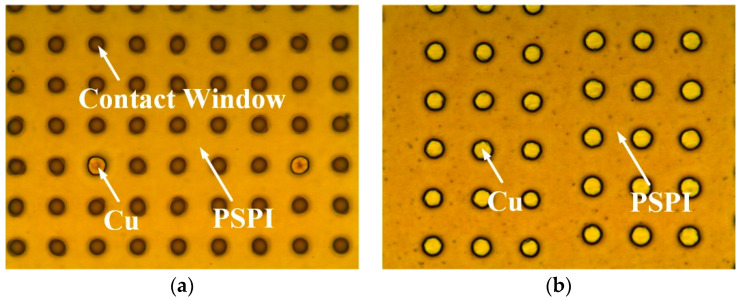
Optical image of patterned contact windows on PSPI after electroplating: (**a**) without plasma bombardment, (**b**) with plasma bombardment.

**Figure 11 micromachines-13-01147-f011:**
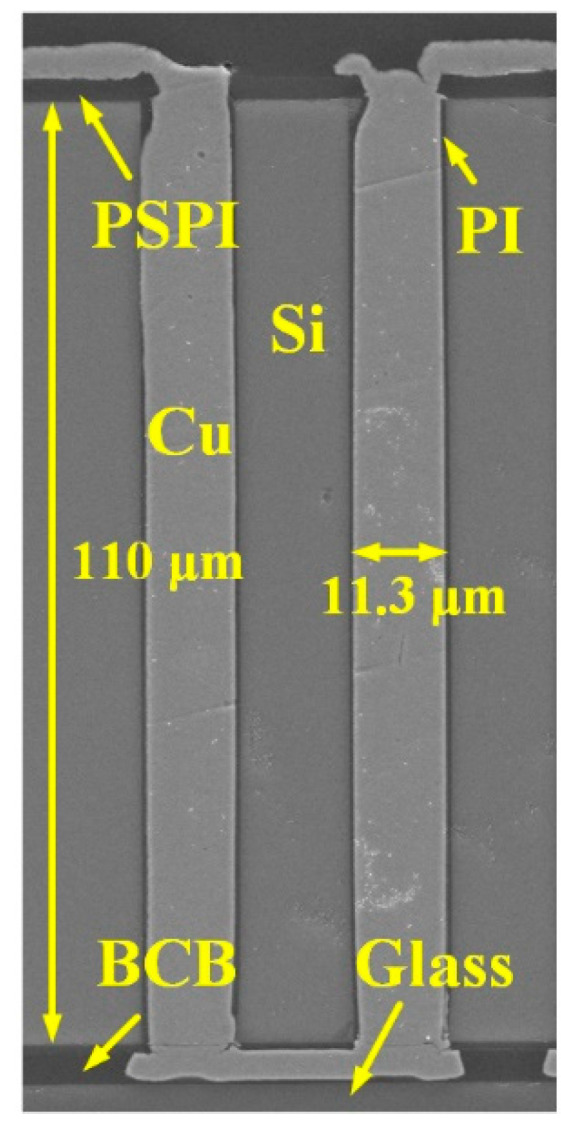
SEM image of fabricated TSVs with double-side RDLs.

**Figure 12 micromachines-13-01147-f012:**
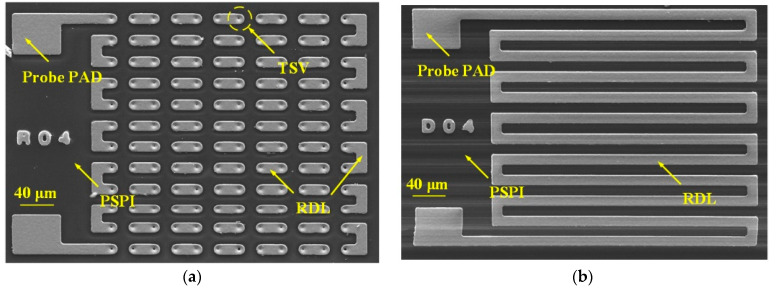
SEM images of the test structures for the DC resistance of the PI-TSV: (**a**) daisy chain structure, (**b**) decoupling structure.

**Figure 13 micromachines-13-01147-f013:**
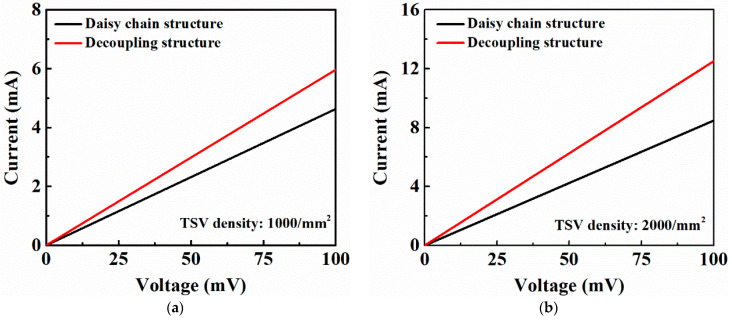
Typical I–V curves of the daisy chain structures and the decoupling structures with: (**a**) TSV density of 1000/mm^2^, (**b**) TSV density of 2000/mm^2^.

**Figure 14 micromachines-13-01147-f014:**
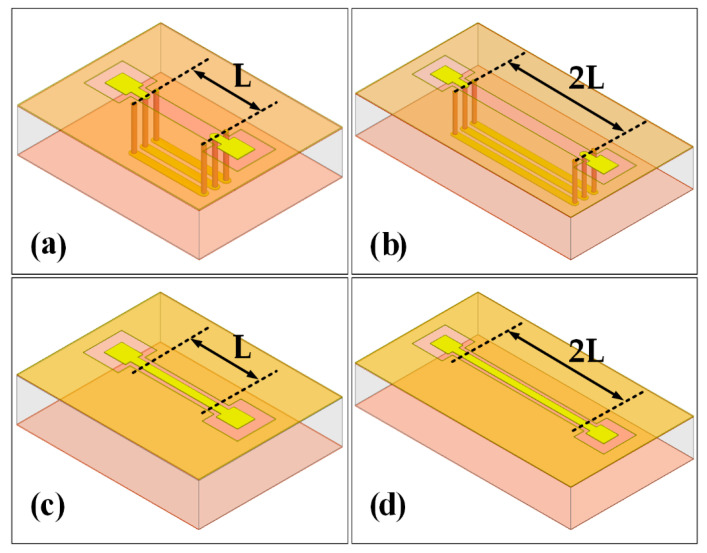
3D models of the L-2L de-embedding structures. (**a**) TSVs and RDLs (L), (**b**) TSVs and RDLs (2L), (**c**) only RDLs (L), (**d**) only RDLs (2L).

**Figure 15 micromachines-13-01147-f015:**
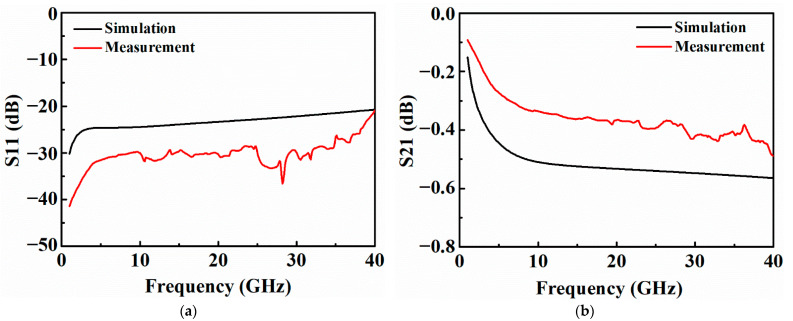
Comparison of the S-parameters of a single TSV between simulation and measurement: (**a**) return loss (S11), (**b**) insertion loss (S21).

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
