# Peer review of "Fabrication and Electrical Characterization of High Aspect Ratio Through-Silicon Vias with Polyimide Liner for 3D Integration"

_micromachines, 2022, doi:10.3390/mi13071147_

Round 1

Reviewer 1 Report

This manuscript demonstrated HAR TSVs with a low-cost process flow. Overall, the process is well documented.

Here are some comments that would help to improve the quality of this manuscript.

1.Since all processes use well-known methods, it is necessary to emphasize the novelty of the process method claimed by the author.

2.Does Fig.7 show the average value after several experiments? A mention of the error rate seems to increase the reliability.

3.In Fig.12, how exactly is the resistance of each structure calculated?

4.In Fi.g13, At the same TSV density, what is the difference in resistance values between the same structures that have measured resistance multiple times?

5.In Fig.12(a) (for daisy chain stucture), the authors specified that 144 TSVs were connected in series via RDLs. However, as a result of direct calculation based on the graph, the values are too far apart.

I think, this part should be carefully confimed.

I hope that the above comments will improve the author's manuscript and make the author's efforts worthwhile in this journal.

Reviewer 2 Report

The authors studied a high aspect ratio TSVs with PI liner. There are already many studies regarding this matter. However, the proposed technique (vacuum assisted spin coating of PI) is new and should be of interest to the reader of this journal. The manuscript is well written. However, there are several issues that have to be addressed prior to publication.

In the introduction, the statement in line 42 is quite strong. There are multiple key technologies for 3D integration, not only the TSV. Others such as wafer bonding (e.g. https://doi.org/10.1016/j.sna.2018.06.021 and http://dx.doi.org/10.1016/j.microrel.2011.04.016) should not be neglected.

Please indicate the corresponding illustration of each steps mentioned in the process design (Fig. 1 and Section 2).

In Section 2.1., it is unclear about what kind of PI is used here. In addition, the explanation of the substrate in use is not sufficient. Can the authors elaborate more about the substrate mentioned in Section 2.1.? Is it as-etched (deep RIE Bosch process) substrate, or did the authors perform some cleaning with chemicals for instance to remove the hydrocarbon at the sidewall of the etched parts? This is an important matter which may influence the PI adhesion on the sidewalls.

Regarding the vacuum assisted spin coating of the PI, can the author elaborate more about Fig. 2b? Did How is the mechanism of this process? Is it the standard process that the entrapped air bubble goes up due to the pressure difference? In addition, to not cause confusion for the readers, the authors should mention that the details of N2 curing step will be mentioned in the following sections.

The steps in Section 2.2. is unclear. Please elaborate about the solutions: conditioner, pre-dip, activator and reducer. Are they commercial solutions or what?

Electroless plating could be formed inside the blind vias even without sputtered Cu layer there. So, what is the role of Cu sputtering in Section 3.2.? What happens if the sputtering process is omitted? The thickness of the Cu seed layer near the surface and the bottom of the via was compared in line 252. However, it seems that the Cu seed layer at near the surface is a combination of sputtered and electroless plated layer.

Please elaborate on the mitigation of the residual stress caused by thinning process in Section 3.3.. How is the condition of the final residual stress after all this process? Any wafer bending? How about after mounting? Since the bumps caused by Cu electroplating often cause some local stresses during the device mounting.

The S-parameters measurement showed a "little" oscillation. What is the mechanism behind the oscillation in the S-parameter measurement?

Round 2

Reviewer 1 Report

Thank you for kind reply.

The author has answered most of my comments well enough.

But still, my comments on the two issues below should be reflected in this manuscript.

1. The authors changed the measurement results in Figure 15 during the revision process.

This can be seen as manipulation of the measurement results.

Also, in Figure 13, the measurement results were also corrected without any explanation.

The measurement result seems to change so easily that it is difficult to trust.

Would you please provide a rationale?

2. The authors claim that the vacuum assisted spin coating method has good consistency.

The authors claim that the vacuum assisted spin coating method has good consistency.

For a statistical comparison with ohter papers mentioned by the author, 

please mention the exact number of 'good consistency' of this work.

For your example, 

at, Shiwei Wang, Yangyang Yan et al. 2016 

Impact of polyimide liner on high-aspect-ratio through-silicon-vias (TSVs): electrical characteristics and copper protrusion

"Achieving at least 30% step coverage"

at, Yangyang Yan et al. 2016

Low capacitance and highly reliable blind through-silicon-vias (TSVs) with vacuum-assisted spin coating of polyimide 

dielectric liners 

"Minimum 30% , Maximum 78% Step Coverage"

at, Miao Xiong et al. 2019

Development of Eccentric Spin Coating of Polymer Liner for Low-Temperature TSV Technology With Ultra-Fine Diameter

"Achieve up to 90% step coverage at a lower cost than adsorption method (ALD)"

Author Response

Thank you very much for your comments and suggestions. Please see the attachment for the detailed responses.

Reviewer 2 Report

The manuscript has been properly revised. It is now more appropriate for publication.

Author Response

Thank you very much for your comments and suggestions.

Round 3

Reviewer 1 Report

Thank you for your reply.

I am now supporting acceptance this manuscript in this journal.